# Nutrition, Bioactive Components, and Hepatoprotective Activity of Fruit Vinegar Produced from Ningxia Wolfberry

**DOI:** 10.3390/molecules27144422

**Published:** 2022-07-11

**Authors:** Yinglei Tian, Ting Xia, Xiao Qiang, Yuxuan Zhao, Shaopeng Li, Yiming Wang, Yu Zheng, Junwei Yu, Jianxin Wang, Min Wang

**Affiliations:** 1State Key Laboratory of Food Nutrition and Safety, Tianjin University of Science and Technology, Tianjin 300457, China; yingleitian@163.com (Y.T.); qiangxiao0907@163.com (X.Q.); yuxuanzhao0206@163.com (Y.Z.); li17627820568@163.com (S.L.); frszzfy@163.com (Y.W.); yuzheng@tust.edu.cn (Y.Z.); 2Ningxia Zhongning Goji Industrial and Innovative Research Institute Co., Ltd., Zhongwei 755000, China; junweiyu@hotmail.com; 3Qiyuantang (Ningxia) Biotechnology Co., Ltd., Zhongwei 755000, China; 4Wisconsin Center for NanoBioSystems, School of Pharmacy, University of Wisconsin, Madison, WI 53705, USA; tkxiaoxin@gmail.com; 5Pharmaceutical Sciences Division, School of Pharmacy, University of Wisconsin, Madison, WI 53705, USA

**Keywords:** wolfberry, vinegar, bioactive ingredients, antioxidant, liver injury

## Abstract

Wolfberry (*Lycium barbarum* L.) is a nutritious and medicinal fruit, and deeply processed products of wolfberry needs to be improved. In this study, nutrition, bioactive compounds, and hepaprotective activity were explored in wolfberry vinegar (WFV). The contents of nutrients including total sugar and protein in WFV samples were 2.46 and 0.27 g/100 mL, respectively. Total phenolic and flavonoid contents in WFV were 2.42 mg GAE/mL and 1.67 mg RE/mL, respectively. *p*-Hydroxybenzoic acid and m-hydroxycinnamic acid were the main polyphenols in WFV. The antioxidant activity of WFV were 20.176 mM Trolox/L (ABTS), 8.614 mM Trolox/L (FRAP), and 26.736 mM Trolox/L (DPPH), respectively. In addition, WFV treatment effectively alleviated liver injury by improving histopathological changes and reducing liver biochemical indexes in CCl4-treated mice. WFV alleviated oxidative damage by inhibiting oxidative levels and increasing antioxidant levels. These results suggest that WFV can be utilized as a functional food to prevent oxidative liver injury.

## 1. Introduction

Recently, the incidence of liver disease worldwide is increasing every year. Hepatitis, cirrhosis, liver failure, and other liver illnesses are all common consequences associated with liver disease, which seriously influences human health [1]. Liver injury is caused by endogenous or exogenous factors [2]. When the body is exposed to harmful stimuli, an excessive amount of reactive oxygen species (ROS) are generated and result in an imbalance between oxidation and antioxidant systems, ultimately leading to oxidative stress [3]. One of the major etiologies of liver injury is oxidative stress. Some plants contain antioxidants, such as polyphenols, polysaccharides, carotenoids, lentinan, vitamin C, and vitamin E. These phytochemicals have antioxidant activity, which can clear free radicals in vivo and conserve the liver health [4]. Therefore, intaking foods with rich antioxidant substances can effectively inhibit the levels of oxidative stress, alleviate liver damage and delay the development of liver disease. 

*Lycium barbarum* L. belongs to the *Solanaceae* family, commonly known as wolfberry, and is an edible and medicinal plant. According to different producing areas, wolfberry can be divided into Ningxia wolfberry and Chinese wolfberry. Among them, Ningxia wolfberry has the largest cultivated area in China. *Lycium barbarum* polysaccharide (LBP), flavonoids, betaine, 18 different amino acids, and 32 different microelements are all found in wolfberry from Zhongning. The contents of LBP and selenium from Zhongning are higher than those from other regions [5]. Therefore, Zhongning wolfberry is recorded in the new Pharmacopoeia of the People’s Republic of China, which has been certified as an “authentic medicinal material” [6]. Recent studies [7] have confirmed that wolfberry with high medicinal value can regulate immune function, blood lipid, and blood glucose, and has effects against tumor, aging, and fatty liver disease. The health benefits of wolfberry have gradually been recognized all over the world. At present, the processing conversion rate of Ningxia fresh wolfberry fruit only reaches 25%, and the varieties of processed and transformed products still need to be supplemented.

Fermented food is processed by beneficial microorganisms and has a unique flavor and a long history. Vinegar is a regular fermented food that comes in two varieties: grain vinegar and fruit vinegar. European countries have combined their own products and dietary habits to produce different kinds of fruit vinegar. Fruit vinegar in Europe is mainly produced by liquid fermentation, and there are several varieties, including balsamic vinegar (Modena), sherry vinegar (Spain), and champagne vinegar (France) [8]. Fruit vinegar is not only rich in small molecular organic acids, phenolic substances, and mineral elements but also contains large molecules such as melanoidins and polysaccharides [9]. It was recently revealed that vinegar lowers blood lipids, and contains antioxidants, preventing liver and cardiovascular diseases, with antibacterial, anti-tumor, and anti-aging effects [10,11]. The functional components derived from raw materials and the fermentation process determine the efficacy of vinegar. Traditional Italian grape vinegar was shown to have superior antioxidant activity to red wines [12,13], and 45% of the antioxidant activity was derived from polyphenols in fruit vinegar [14]. Owing to the anti-inflammatory and antioxidant character, Beh et al. [15] found that 2 mL/kg of nipa vinegar protected the liver induced by paracetamol in mice. Bouazza et al. [16] reported that three fruit vinegars (pomegranate vinegar, prickly vinegar, and apple vinegar) significantly regulated lipid metabolism and reduced liver injury in hyperlipidemia mice. Wolfberry is fermented to produce wolfberry fruit vinegar (WFV) by microbial processes, which can extend the shelf life of wolfberry, and improve its nutritional and functional value. However, the functional components of WFV and its hepatoprotective activity are still unclear.

In this study, WFV was produced by alcohol and acetic acid fermentation. Firstly, the physicochemical and nutritional ingredients were determined in WFV. In addition, the bioactive ingredients and antioxidant activity were detected. Moreover, the positive influence of WFV against CCl_4_-treated liver trauma in mice was investigated. The finding will provide fermented and healthy foods of wolfberry to prevent liver disease.

## 2. Results and Discussion

### 2.1. Physicochemical Indices and Nutrients in WFV

The physicochemical indices and nutrients in WFV samples were detected and shown in Table 1. The pH value of traditional cereal vinegar and fruit vinegar was 2.0–3.5 [17], and the pH value of WFV was in this range. The Chinese national standard (GB/T 18623-2011) states that the total acid content of vinegar should be greater than 4.5 g/100 mL. The total acid of WFV was 6.72 ± 0.12 g/100 mL, which satisfied this standard. It has been reported that non-volatile acids including lactic acid, malic acid, and citric acid can moderately increase soft acidity and reduce the irritability of smelling and drinking [18]. In this investigation, the content of non-volatile acid was 0.84 ± 0.11 g/100 mL, accounting for about 12.5% of the total acid. In addition, the contents of soluble solids, and amino nitrogen are both important indicators of vinegar nutrition, flavor, and fermentation degree [19]. The percentage of soluble solids was 8.66 ± 0.32%, and the content of amino nitrogen was 0.07 ± 0.01 g/100 mL in WFV. The alcohol level of the fermentation broth decreased during acetic acid fermentation. When the fermentation was near the end, the reducing sugar precursor substances were consumed in large quantities, increasing the reducing sugar content. At the end of fermentation, the ethanol was almost depleted and the alcohol degree reached a low level [20]. In this study, the alcohol degree of WFV was 0.06%(vol), and the content of reducing sugar was 1.42 ± 0.14 g/100 mL. The higher the reducing sugar and soluble solids content is, the higher the content of nutrients in vinegar [21]. The soluble solids were all compounds that can be dissolved in liquids, including sugars, acids, vitamins, and minerals, and which were important indicators for evaluating the quality of vinegar. The contents of nutrients including total sugar and protein in WFV were 2.46 ± 0.14 and 0.27 ± 0.01 g/100 mL, respectively. The fat content was very low. The data mentioned above suggested that WFV was a healthy low-sugar and low-fat food. And then the bioactive ingredients in WFV samples were investigated to further explore its functional properties.

### 2.2. Bioactive Ingredients in WFV

Wolfberry is rich in bioactivity substances such as polysaccharides, betaine, carotenoid, phenolics, and flavonoids [22]. Organic acids, amino acids, polyphenols, and flavonoids, which are bioactive components in vinegar, are derived mainly from raw materials and manufacturing methods [23]. In this study, bioactive ingredients were detected in WFV samples (Table 2). Total phenolic content (TPC) and total flavonoid content (TFC) in WFV samples were 2.42 ± 0.05 mg GAE/mL and 1.67 ± 0.03 mg RE/mL, respectively. Liu et al. [23] reported the TPC (0.03–3.22 mg GAE/mL) and TFC (0.005–0.75 mg QE/mL) in 23 kinds of fruit vinegar. The TPC in Balsamic vinegar of Modena was the highest, and the TFC in Aceto Balsamico di Modena was higher than those in other kinds of vinegar. Our previous study [24] showed that the TPC of Shanxi aged vinegar with different aging years varied from 1.43 to 3.73 mg GAE/mL, while the TFC was 1.97–3.43 mg RE/mL. Our data indicated that the TPC in WFV was at a higher level when compared with that in fruit vinegar and grain vinegar. However, when compared with the above-mentioned Shanxi aged vinegar, the TFC in WFV was considered to be at a high level. In addition, the water-soluble polysaccharide—LBP—is the main bioactive component in wolfberry, which represents the quality of wolfberry [25]. Carotenoids also exist in wolfberry, which is beneficial to the human eyes [26]. The LBP content in WFV was 8.94 ± 0.27 mg/mL, and the betaine and carotenoid levels in WFV were 2.88 ± 0.22 and 0.42 ± 0.02 mg/mL, respectively. Wang et al. [27] found that the betaine content (>0.59 mg/mL) in Chinese cereal vinegar was higher than that in European grape vinegar, which was less than the betaine content in WFV. Ali et al. [28] found that the carotenoid contents in red and blackened jujube vinegar were 0.45–1.00 and 1.48–3.47 mg/100 mL, which was much lower than the vinegar we produced. The bright red color of wolfberry was also attributed to high levels of carotenoids, and the carotenoid content in dried Ningxia goji berries was 3.39 mg/100 g [29]. Our data indicated that more carotenoids were preserved in the production process of WFV. In the present study, the results showed that LBP, betaine and TPC were the main bioactive components. Numerous studies have demonstrated that LBP, betaine and polyphenols are positively correlated with antioxidant activity [30,31,32]. The antioxidant activities of WFV samples were further examined in the next experiment.

### 2.3. The Antioxidant Activity of WFV

The antioxidant activity in WFV was detected by 2,2-azino-bis(3-ethylbenzothiazoline-6-sulphonic) acid (ABTS) and 2,2-diphenyl-1-picrylhydrazyl (DPPH) (Figure 1). As displayed in Figure 1, the antioxidant activities of WFV were 20.842 ± 0.644 mM Trolox/L (ABTS), and 26.736 ± 1.238 mM Trolox/L (DPPH), respectively. Kelebek et al. [33] reported eight different brands of apple cider vinegar and grape vinegar from Turkish market. The highest antioxidant activity of apple cider vinegar was 20.19 ± 0.41 mM Trolox/L (ABTS) and 14.69 ± 0.30 mM Trolox/L (DPPH), respectively. The highest antioxidant activity of grape vinegar was 17.96 ± 1.34 mM Trolox/L (ABTS) and 14.43 ± 0.97 mM Trolox/L (DPPH), respectively. By comparison, the ABTS of WFV in this study was slightly higher than that of apple cider vinegar (*p* > 0.05), and significantly higher than that of grape vinegar (*p* < 0.05). In addition, the DPPH value of WFV was significantly higher than that of two commercial fruit vinegars (*p* < 0.001). It has been reported that various fruits contain different bioactive substances such as polyphenols and vitamins, providing antioxidant activity which influences the properties of vinegar [34]. In addition, new bioactive compounds such as organic acids and polyphenols can be produced during the fermentation process [35,36]. Some studies have demonstrated that the antioxidant activities of fruit vinegars were related to the raw materials and manufacturing techniques [37,38,39]. Generally, the results showed that the antioxidant capability of WFV was superior to that of apple cider and grape vinegar, which was due to different raw materials and manufacturing techniques. Additionally, the polyphenolics of antioxidant substances in WFV samples were investigated in further experiments. It has been reported that the polyphenolics in vinegars are mainly derived from raw materials, which are closely related to the antioxidant activity of vinegars [23]. Therefore, the polyphenolic compounds in WFV samples were investigated in a further experiment.

### 2.4. Analysis of Polyphenols in WFV

Furthermore, the kinds and contents of polyphenol in WFV samples were detected by GC-MS. Eighteen kinds of polyphenolic compounds were detected in WFV samples (Table 3). It was found that *p*-hydroxybenzoic acid (0.317 mg/mL) and m-hydroxycinnamic acid (0.311 mg/mL) were the top two polyphenolic compounds in WFV, followed by 3-(4-hydroxy-3-methoxyphenyl) propionic acid, chlorogenic acid, and 2-(4-hydroxyphenyl) ethanol. The above five polyphenols accounted for 55.83% of the total polyphenol content. It had been reported that 2-(4-hydroxyphenyl) ethanol is produced by tyrosine metabolism in yeasts during the alcoholic fermentation stage [40]. *p*-hydroxybenzoic acid is mainly obtained from the oxidation of 2-(4-hydroxyphenyl) ethanol in the process of acetic acid fermentation [41]. It was found that 3-(4-hydroxy-3-methoxyphenyl) propionic acid has anti-oxidative activity, which was related to anti-inflammatory effects and regulation of lipid metabolism [42]. Chlorogenic acid, as a polyphenol monomer, was as abundant in vinegar as other fruit vinegar [10]. Alonso et al. [43] found that the sherry vinegar aged in the wood had the highest content of polyphenols, including tyrosol (97.96 mg/mL), gallic acid (95.01 mg/mL), catechin (60.02 mg/mL), and *p*-hydroxybenzoic acid (33.47 mg/mL). According to the study, the composition of vinegar was linked to the raw materials used and the manufacturing process. Nakamura et al. [44] compared high-Brix apple vinegar (HBAV) with regular apple vinegar (RAV). It was found that chlorogenic acid (19.6 mg/100 mL), 4-p-coumarylquinic acid (13.5 mg/100 mL) and caffeic acid (0.76 mg/100 mL) were the three main phenolic substances. The content of chlorogenic acid and its isomer accounted for 56.9% of the total phenolic content and contributed 41.7% to the superoxide dismutase activity of vinegar. Chlorogenic acid was measured as a phenolic with a content of 3.1 mg/100 mL in RAV. It was demonstrated that these phenols in vinegar were derived from raw materials and retained through acetic acid fermentation [23]. Collectively, *p*-hydroxybenzoic acid, m-hydroxycinnamic acid, and 3-(4-hydroxy-3-methoxyphenyl) propionic acid were the main polyphenols in WFV, which were mainly derived from the fermentation period and raw wolfberry. Moreover, the hepatoprotective effect of WFV was evaluated in vivo.

### 2.5. Effect of WFV on the Damaged Liver in Mice

The effect of WFV on oxidative liver injury was investigated by observing the changes in liver tissue morphology in mice (Figure 2). In Figure 2A, the livers of mice in WFV group were similar to that in the control group, showing a normal dark red color. The liver appearance in the model group was diffusely enlarged, and the color was lighter and whiter than that in the control group. On the contrary, the hepatic tissue in the CCl_4_+silybin group was not swollen, and the color was darker than that of the model group. When compared to the model group, the hepatic tissue condition in the CCl_4_+WFV group was closer to that in the CCl_4_+silybin group, indicating the positive effect. In addition, the basic morphology of cells in the liver tissue was observed by H&E. (Figure 2B). As shown in Figure 2B, the liver cells in the WFV group were without the presence of steatosis, which was similar to that in the control group. Generally, there are no obvious anatomical and histopathological changes between the control group and WFV group. The results indicate that WFV has no cytotoxic effect on normal cells. Hepatocytes in the model group were accompanied by partial inflammation, steatosis, and swelling. The hepatic cells in the CCl_4_+WFV group and the CCl_4_+silybin group were characterized by less steatosis and inflammation, indicating alleviation of hepatic tissue injury. As shown in Figure 2C, the steatosis scores between control and WFV groups were not significantly altered. The steatosis score in the model group was significantly higher than that in the control group. On the contrary, the level of liver steatosis in the CCl_4_+WFV group was significantly decreased when compared with that in the model group, which suggested that the degree of hepatic steatosis was alleviated by WFV pretreatment. Previous studies [45,46] reported that wolfberry had a protective effect on oxidative liver injury. Omar et al. [47] found that apple cider vinegar protected liver injury resulting from nicotine toxicity by reducing liver biochemical markers and the size of the hepatocytes’ nuclei. In general, the findings, as well as previous references, show that WFV can reverse the damaged hepatocytes and reduce liver trauma through morphology examination. Next, some biochemical indexes were further determined to verify the hepatoprotective effect of WFV.

### 2.6. Effect of WFV on Biochemical Indexes in Mice with Oxidative Liver Injury

Figure 3 depicts the hepatoprotective effect of WFV on serum and hepatic ALT and AST activity. ALT and AST are two kinds of transaminase in cytoplasm and mitochondria, which can be detected in hepatic tissue and serum. Both of them are the most commonly used indicators of liver function examination [48]. In the model group, both ALT and AST activities were significantly increased in serum and hepatic tissue. However, the boosted levels of ALT and AST were considerably declined (*p* < 0.05) in CCl_4_+WFV groups, and similar results were shown in CCl_4_+silybin groups. The data indicated that the WFV-relieved injury to the liver was proved by the histopathological observation mentioned above. Both TC and TG can be biosynthesized and stored in the liver. Liver injury can decrease the metabolic capacity of the liver, and cause abnormal lipid metabolism [49]. TG and TC in serum also had a similar trend in the CCl_4_+WFV group, suggesting the reduction of liver injury. In addition, AKP in hepatocytes is tightly bound to the hepatocyte membrane, and the activity of AKP increases when bile metabolism is abnormal [50]. LDH is a kind of hydrogen transfer enzyme abundant in liver tissue [51]. The results verified that CCl_4_ significantly increased the levels of AKP and LDH in the hepatic tissue, while silybin significantly decreased these levels (*p* < 0.01). The activity of AKP and LDH in the CCl_4_+WFV group was lower than those in the model group (*p* < 0.01). It had been reported that LBP and wolfberry juice were used to act as an antidote by activating related detoxification enzymes (cytochrome P450, UDP-glucuronosyltransferase, and glutathione S-transferase), which was also attributed to the bioactive substances contained in it [52]. It was found that apple polyphenols significantly reduced the increase of liver functional enzymes (ALT and AST), and reduced the degree of liver injury, which was attributed to the scavenging effect of free radicals and antioxidant activity from apple polyphenols [53]. The results mentioned above suggested that WFV decreased the levels of liver function indexes, and had a positive effect on liver injury. Subsequently, the oxidative and antioxidative paraments were evaluated to explore the mechanism of the hepatoprotective activity of WFV.

### 2.7. Effect of WFV on Oxidative and Antioxidative Levels in Mice

The positive effect of WFV on oxidative liver trauma was studied by measuring the levels of oxidative and antioxidative indexes in the liver of mice. Oxidative stress causes damage to cells, including lipids and DNA, which induce the activation of macrophages in the extracellular matrix, resulting in tissue destruction and inflammation [54]. As shown in Figure 4A–D, CCl_4_ accelerated ROS generation and significantly increased the contents of MDA, 4-HNE, and 8-OHdG in the liver (*p* < 0.01). 4-HNE is an intermediate product of lipid oxidation, and the end product is MDA [55]. 8-OHDG is one kind of important indicator of oxidative injury to DNA caused by endogenous or exogenous factors [56]. The elevated levels of these three indexes can cause liver cell toxicity and even apoptosis, and eventually lead to tissue lesions. In contrast, silybin treatment significantly lowered the levels of these oxidative products (*p* < 0.01). These oxidative products in the CCl_4_+WFV group were significantly decreased when compared with the model group (*p* < 0.05). These results suggested that the inhibitory effect of WFV on oxidative levels in mice was similar to that of the CCl_4_+silybin group.

The antioxidant system, including SOD, CAT, GSH, and GSH-Px maintains the oxidative balance of the organism [57]. SOD and CAT are vital antioxidant enzymes in the body, and GSH is the substrate for GSH-Px to decompose hydrogen peroxide [58]. Previous studies [59,60] indicated that CCl_4_-treated liver injury in mice can reduce SOD, CAT, GSH, and GSH-Px levels in the hepatic tissue, which is basically consistent with the oxidation liver injury model established in this experiment. As shown in Figure 4E–H, the activities of the antioxidant enzymes (CAT, SOD, and GSH-Px) and GSH content in the hepatic tissue were significantly decreased after CCl_4_ treatment (*p* < 0.05). Silybin administration significantly improved these antioxidant indexes (*p* < 0.05). The levels of antioxidant indexes in CCl_4_+WFV group were similar to those in the silybin group, which had a good positive effect on liver injury. Collectively, WFV had a role in balancing the levels of oxidation and antioxidation, which alleviated oxidative damage in the liver.

## 3. Materials and Methods

### 3.1. Chemical Reagents

Anhydrous ethanol, phenol, sulfuric acid, n-hexane, propanone, methylbenzene, KOH, H_2_SO_4_, ether, ethyl acetate, were supplied by Sinopharm Chemical Reagent Co., Ltd. (Shanghai, China). Methanol, 2,2-diphenyl-1-picrylhydrazyl (DPPH), and 2,4,6-trihydroxybenzoic acid, were obtained from Sigma-Aldrich (St. Louis, MO, USA). A total antioxidant activity assay kit (2,2-azino-bis(3-ethylbenzothiazoline-6-sulphonic) acid (ABTS)) was garnered from Beyotime Biotechnology (Shanghai, China).

### 3.2. Production of WFV

The fresh wolfberry was obtained from Zhongning County (Ningxia, China), and the wolfberry juice was processed by simply crushing the fresh wolfberry. Then, 0.02% pectinase was added to the wolfberry juice. The wolfberry juice was maintained in a 45 °C water bath for 4 h before being heated (85 °C, 10 min) to lower the enzyme’s activity. Initial sugar content of wolfberry juice was adjusted to 20° Bx with white granulated sugar. For the alcoholic fermentation stage, 0.02% Active Dry Wine Yeast RW (Angel Yeast., Yichang, Hubei, China) was added to the wolfberry juice and then cultivated in an incubator at 22 °C for 5 d. Next, wolfberry wine was centrifuged and inoculated with the activated *Acetobacter Pasteurianus* CGMCC 3089 (10%, *v*/*v*) at 30 °C and 180 rpm for 6 d. After fermentation, WFV was centrifuged to collect the supernatant. Three batches of WFV were reserved at 4 °C for subsequent analysis. All three batches of the wolfberry fruits were collected in the same year to produce WFV in order to reduce batch-to-batch variation.

### 3.3. Determination of Nutrients and Physicochemical Indices in WFV

The contents of total sugar (GB/T15038-2006), fat (GB 5009.6-2016), protein (GB 5009.5-2016), total acid (GB/T 12456-2021), involatile acid (GB 5009.235-2016), amino nitrogen (GB/T 5009.157-2016), and reducing sugar (GB5009.7-2016) were detected according to National Standards. The content of soluble solids was determined by an LH-T20 hand-held refractometer (Hangzhou Lohand Biological Co., Ltd., Hangzhou, China). The pH values were measured by an SG78-SevenGo Duo pro ™ pH meter (METTLER TOLEDO, Zurich, Switzerland). 

The WFV samples were diluted and filtrated through a 0.22 μm membrane for the next steps. The alcohol content of WFV was analyzed by high-performance liquid chromatography (HPLC) (Agilent, Santa Clara, CA, USA). The detector was a refractive index detector, the chromatographic column was an Aminex HPX-87H column (Bio-Rad, Hercules, CA, USA, 300 × 7.8 mm) and column temperature was 30 °C. The mobile phase was 5 mmol/mL H_2_SO_4_-ultrapure water flowing at 0.6 mL/min.

### 3.4. Determination of Active Ingredients in WFV

#### 3.4.1. TPC and TFC

TPC and TFC assays of WFV were measured in line with the methods utilized before [24]. The calibration curves of TPC and TFC were used with gallic acid equivalents (mg GAE/mL) and rutin equivalents (mg RE/mL) as content units, respectively.

#### 3.4.2. LBP

The WFV samples were concentrated at 40 °C, and four times the volume of anhydrous ethanol was added and stood for 12 h. Then, the mixture was centrifugated (8000 r/min, 10 min). The precipitation was gathered and deionized water was added to redissolve the crude polysaccharide. The protein removing reagent—sevage reagent (Dichloromethane:1-Butanol = 4:1)—was added and centrifuged (8000 r/min, 10 min) and the supernatant was obtained. We repeated the above steps until the protein was completely removed. The supernatant was in dialysis for 72 h in an 8–14 kDa dialysis bag (Beijing, China) and collected.

The polysaccharide contents were detected using the phenol–sulfuric acid method. Amounts of 1 mL diluted extraction and 1 mL deionized water were mixed. Amounts of l mL 5% phenol solution and 5 mL H_2_SO_4_ were added, and then incubated for 10 min. The mixture was heated in a 40 °C water bath for 15 min. The value of A_490_ was detected and transformed to mg/mL.

#### 3.4.3. Carotenoids

WFV samples (0.2 mL), 20 mL mixture (n-hexane: propanone: ethanol: methylbenzene = 10:7:6:7), and 1 mL 40% KOH methanol solution were mixed. The mixture was soaked overnight at 35 °C, and then 40 mL ether and 100 mL deionized water were mixed. We repeated the operation at least twice again. After absorbing water by anhydrous sodium sulfate, the samples were collected and diluted to 100 mL using ether. The value of A_450_ was detected. The computational formula was as follows:The carotenoid content (mg/100 mL) = (A × V_2_ × 1000)/(V_1_ × E)(1)
where A means the absorbance of WFV samples, V_1_ means the volume of WFV samples, V_2_ means 100 mL and E means the absorbance of carotenoid (2500).

#### 3.4.4. Betaine

0.2 mL WFV samples and 1 mL 80% methanol were blended and heated in a water bath (60 °C, 30 min). After that, the blending was centrifuged (10,000 rpm, 15 min). Then, we heated the liquid supernatant in a 70 °C water bath to volatilize methanol completely. We mixed the sample with deionized water to 1 mL, and the betaine extract was obtained. The betaine content was determined at 525 nm by the betaine content assay kit (Beijing solarbio science & technology., Beijing, China).

#### 3.4.5. Polyphenols

20 mL WFV sample was centrifuged (8000 r/min, 15 min) for stand-by. The supernatant was filtered by a 10 kDa ultrafiltration cup (Shanghai, China) and purified through a column containing AB-8 macroporous resin (Beijing, China), and eluted first with 200 mL deionized water, followed by elution with ethanol. The final eluent was concentrated and subsequently redissolved with deionized water. The polyphenols were extracted with ethyl acetate and concentrated. Finally, 2 mL BSTFA + 1% TMCS was added and reacted in water bath (70 °C, 4 h).

Polyphenols in WFV samples were identified and quantified by GC-MS (Shimadzu, Japan). The chromatographic column was an Rtx-5MS (Shimadzu, 30 m × 0.25 mm × 0.25 μm). Inlet temperature (300 °C), helium gas flow rate (1 mL/min), and split ratio (10) were the detection conditions. The temperature of heating program was 80 °C for 2 min firstly, and then 315 °C at 5 °C/min speed. The ion source temperature was 200 °C, interface temperature was 220 °C, solvent delay time was 3 min, and we scanned from 35 to 1000 *m*/*z*. The polyphenol compounds were identified by the mass-to-charge ratio and retention time based on the mass spectrometry library of the National Institute of Standards and Technology (NIST). The matching degrees of polyphenols compounds (≥) 80% were selected and identified. Meanwhile, 2,4,6-trihydroxybenzoic acid was used as an internal standard substance. The content of polyphenols in WFV was calculated by comparing the peak area ratio of the predicted component to that of the internal standard.

### 3.5. Antioxidant Activity of WFV

Antioxidant activities of WFV samples were measured by ABTS and DPPH assays. ABTS radical scavenging capacity was measured through the instructions of total antioxidant capacity assay kits. The DPPH free radical scavenging ability of WFV was evaluated in line with the previous measure [61] with some alterations. The concentration of DPPH working solution was modified to 0.1 mmol/L. The rest of the steps were the same. The results were expressed as equivalent concentrations of Trolox in mM.

### 3.6. Animals and Diets

All animal experiments were carried out in line with the guidelines of the institutional animal ethics committee. Forty ICR mice (5–6 weeks old, 18–22 g) were placed at a constant temperature (22 ± 2 °C, 55 ± 5% humidity) and adaptively fed for one week. The mice were randomly grouped: control group; WFV (2.5 mL/kg b.w.) group; model group (CCl_4_, 1 mL/kg b.w.); CCl_4_+silybin (100 mg/kg b.w.) group; CCl_4_ + WFV (2.5 mL/kg b.w.) group. The CCl_4_+silybin and CCl_4_ + WFV groups were separately fed with oral treatments once a day for 14 d. On the fourteenth day, Group III, IV, V were given an intraperitoneal injection of CCl_4_ after administration for 2 h. The mice mentioned above were sacrificed 12 h later. The blood samples and hepatic tissues were gathered and frozen in refrigerator for subsequent experiments.

### 3.7. Biochemical Analysis

The values of aminotransferase (ALT), aspartate aminotransferase (AST), total cholesterol (TC), and triglyceride (TG) in the serum and the values of ALT, AST, alkaline protease (AKP), lactate dehydrogenase (LDH), superoxide dismutase (SOD), catalase (CAT), reduced glutathione (GSH), glutathione peroxidase (GSH-Px), and malonaldehyde (MDA) in hepatic tissues were measured by assay kits (Nanjing Jiancheng Bioengineering Institute, Nanjing, China). The ELISA kits (San Diego, CA, USA) were followed to detect the activities of reactive oxygen species (ROS), 8-hydroxydeoxyguanosine (8-OHdG), and 4-hydroxynonenal (4-HNE) in the liver tissues.

### 3.8. Histopathological Examination

The fresh hepatic tissues were immobilized with 4% paraformaldehyde for 24 h. The tissues were dewatered with alcohol, immersed in wax, embedded, and sliced. After gradient elution, the samples were dyed with hematoxylin–eosin (H&E). Finally, the slices were examined using an optical microscope (Nikon, Tokyo, Japan).

### 3.9. Statistical Analysis

The data were analyzed using GraphPad Prism 8.0.1 software and represented as means and standard deviations (SD). A one-way analysis of variance (ANOVA) for data was performed using SPSS 26.0 software. Statistical significance was defined as a value of *p* < 0.05.

## 4. Conclusions

In the present study, WFV made from wolfberry had a superior total acid content, which was in accordance with the standard of vinegar. The total sugar and protein in WFV samples were 2.46 ± 0.14 and 0.27 ± 0.01 g/100 mL respectively. In addition, the bioactive components included polyphenols, flavonoids, LBP, and betaine. The antioxidant activity of WFV by ABTS, FRAP, and DPPH were 20.176 ± 0.154, 8.614 ± 0.079, and 26.736 ± 0.242 mM Trolox/L, respectively. The main polyphenols were *p*-hydroxybenzoic acid, m-hydroxycinnamic acid, 3-(4-hydroxy-3-methoxyphenyl) propionic acid, chlorogenic acid, and 2-(4-hydroxyphenyl) ethanol. In vivo, the results showed that WFV ameliorated liver tissue injury and reduced liver function markers in CCl_4_-treated mice. WFV exhibited a hepaprotective role by improving antioxidant levels and lowering oxidative levels. In conclusion, WFV can be utilized as a functional and anti-hepatic injury food.

## Figures and Tables

**Figure 1 molecules-27-04422-f001:**
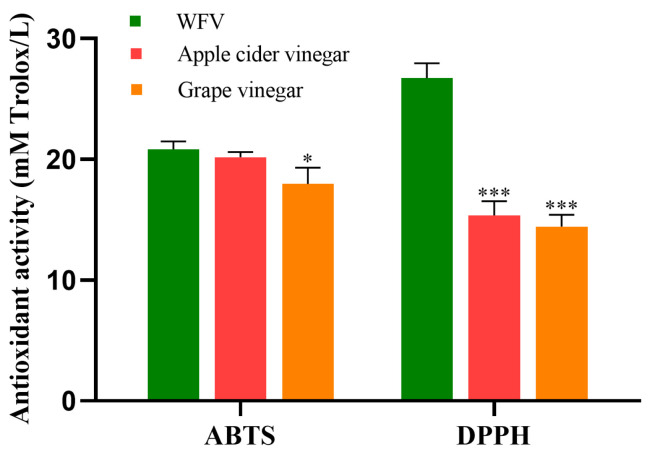
The antioxidant capacity of WFV was measured by ABTS, DPPH assays. The antioxidant capacities of apple cider vinegar, and grape vinegar were cited from Ref. [33]. * *p* < 0.05, *** *p* < 0.001 vs. WFV.

**Figure 2 molecules-27-04422-f002:**
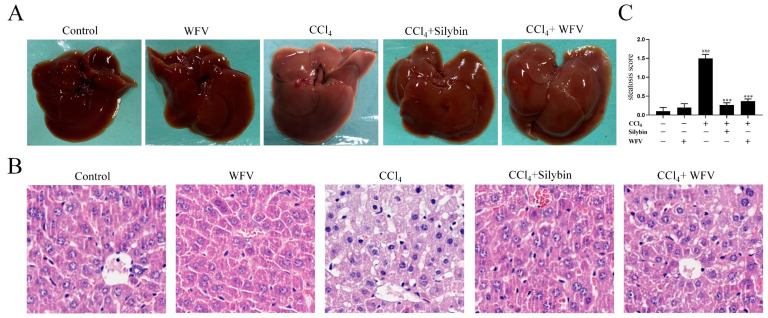
Effect of WFV on the damaged liver in mice. (**A**) Anatomical examination of liver in mice. (**B**) Hepatic tissues were dyed with H&E (200× magnification). (**C**) Steatosis score of liver in mice. ^###^ *p* < 0.001 vs. control group. *** *p* < 0.001 vs. model group.

**Figure 3 molecules-27-04422-f003:**
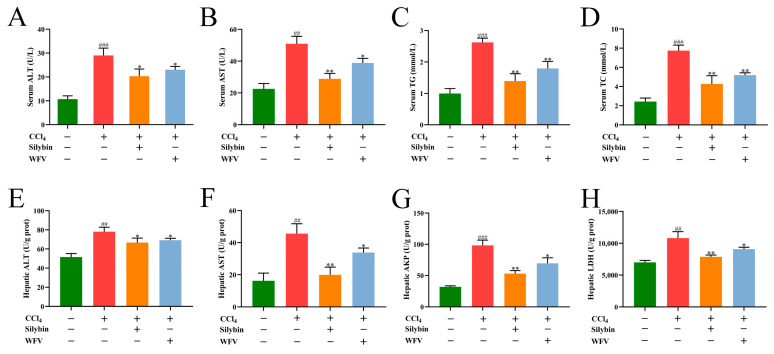
Effect of WFV on biochemical indexes in mice with oxidative liver trauma. The levels of (**A**) ALT, (**B**) AST, (**C**) TG, and (**D**) TC were detected in the serum. The levels of (**E**) ALT, (**F**) AST, (**G**) AKP, and (**H**) LDH were detected in the liver. ^##^ *p* < 0.01, ^###^ *p* < 0.001 vs. control group. * *p* < 0.05, ** *p* < 0.01 vs. model group.

**Figure 4 molecules-27-04422-f004:**
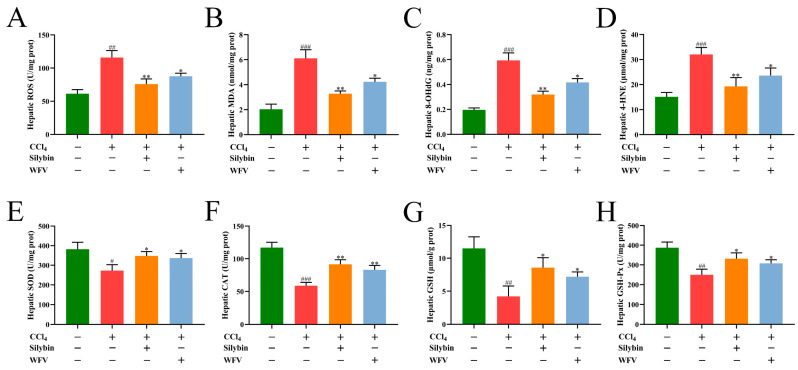
Effect of WFV on oxidative and antioxidative levels in mice with liver injury. Hepatic levels of ROS (**A**), MDA (**B**), 8-OHdG (**C**), 4-HNE (**D**), SOD (**E**), CAT (**F**), GSH (**G**), and GSH-Px (**H**) were detected by a microplate reader. ^#^ *p* < 0.05, ^##^ *p* < 0.01, ^###^ *p* < 0.001 vs. control group. * *p* < 0.05, ** *p* < 0.01 vs. model group.

**Table 1 molecules-27-04422-t001:** Physicochemical indices and nutrients in WFV.

	Parameters (Units)	Contents
Physicochemical indices	pH value	3.38 ± 0.08
	Total acid (g/100 mL)	6.72 ± 0.12
	Non-volatile acid (g/100 mL)	0.84 ± 0.11
	Soluble solids (%)	8.66 ± 0.32
	Amino nitrogen (g/100 mL)	0.07 ± 0.01
	Reducing sugar (g/100 mL)	1.42 ± 0.14
	Alcohol (% vol)	0.06 ± 0.01
Nutrients	Total sugar (g/100 mL)	2.46 ± 0.14
	Protein (g/100 mL)	0.27 ± 0.01
	Fat (g/100 mL)	0.12 ± 0.02

**Table 2 molecules-27-04422-t002:** Content of functional component and antioxidant activity in WFV.

Indexes	TPC (mg GAE/mL)	TFC (mg RE/mL)	LBP (mg/mL)	Betaine (mg/mL)	Carotenoids (mg/mL)
WFV	2.42 ± 0.05	1.67 ± 0.03	8.94 ± 0.27	2.88 ± 0.22	0.42 ± 0.02

TPC: total phenolic content; TFC: total flavonoid content; LBP: *Lycium barbarum* polysaccharide.

**Table 3 molecules-27-04422-t003:** GC-MS analysis of polyphenols in WFV.

	Polyphenols	Retention Time (min)	Content (mg/mL)
			WFV
1	*p*-hydroxybenzaldehyde	14.602	0.022 ± 0.001
2	salicylic acid	18.345	0.059 ± 0.002
3	2-(4-hydroxyphenyl) ethanol	19.817	0.216 ± 0.024
4	*p*-hydroxybenzoic acid	21.146	0.317 ± 0.011
5	*p*-hloroglucinol	21.691	0.105 ± 0.004
6	4-hydroxyphenylacetic acid	24.220	0.028 ± 0.001
7	vanillic acid	24.360	0.097 ± 0.003
8	m-hydroxycinnamic acid	24.880	0.311 ± 0.011
9	protocatechuic acid	25.665	0.022 ± 0.001
10	3-(4-hydroxy-3-methoxyphenyl) propanoic acid	27.190	0.151 ± 0.005
11	3-(4-hydroxyphenyl)-2-hydroxypropionic acid	27.450	0.038 ± 0.001
12	3-hydroxy-4-methoxycinnamic acid	27.754	0.062 ± 0.002
13	4-hydroxycinnamic acid	28.060	0.131 ± 0.005
14	3,4-diphenylpropionic acid	28.315	0.027 ± 0.001
15	3-(4-hydroxy-3-methoxyphenyl) propionic acid	30.625	0.222 ± 0.008
16	ferulic acid	31.130	0.134 ± 0.005
17	caffeic acid	32.025	0.143 ± 0.005
18	chlorogenic acid	47.890	0.222 ± 0.008

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
