# Peer review of "Nutrition, Bioactive Components, and Hepatoprotective Activity of Fruit Vinegar Produced from Ningxia Wolfberry"

_molecules, 2022, doi:10.3390/molecules27144422_

Round 1

Reviewer 1 Report

Add legend in table 2 with the meaning of the abbreviations.

Clarify how polyphenols were identified by GC-MS. Did the authors use authentic samples of the identified compounds? Literature data?

There is a lack of a general discussion linking the results of each experiment and the chemical analysis. The results were described, but not related to each other.

Author Response

Response to Reviewer 1 Comments

Point 1: Add legend in table 2 with the meaning of the abbreviations.

Response 1: Thanks for your kindly reminding. As suggested, the legend with the meaning of the abbreviations has been added in table 2. The related content was added as follows: TPC: total phenolic content; TFC: total flavonoid content; LBP: Lycium barbarum polysaccharide.

The related content was added in table 2 (Page 4 Line 143).

Point 2: Clarify how polyphenols were identified by GC-MS. Did the authors use authentic samples of the identified compounds? Literature data?

Response 2: Thanks for your kindly suggestion. We are sorry for the unclear description. In this study, polyphenols from WFV samples were detected by GC-MS. The polyphenols compounds were identified by the mass-to-charge ratio and retention time based on the mass spectrometry library of the National Institute of Standards and Technology (NIST). The matching degrees of polyphenols compounds (≥) 80% were selected and identified. Meanwhile, the 2,4,6-trihydroxybenzoic acid was used as an internal standard substance. The content of polyphenols in WFV was calculated by comparing the peak area ratio of the predicted component to that of the internal standard.

In this study, we used WFV samples to detect polyphenols. 18 kinds of polyphenolic compounds were detected in WFV samples. It was found that p-hydroxybenzoic acid (0.317mg/mL) and m-hydroxycinnamic acid (0.311 mg/mL) were the top two polyphenolic compounds in WFV, followed by 3-(4-Hydroxy-3-methoxyphenyl) propionic acid, chlorogenic acid, and 2-(4-hydroxyphenyl) ethanol. The above five polyphenols accounted for 55.83% of the total polyphenol content.   

As suggested, the related contents were added in the Results and discussion section 2.4 (Page 5 Line 174-178), and 3.4.5 Materials and Methods section (Page11 Line 378-384).

Point 3: There is a lack of a general discussion linking the results of each experiment and the chemical analysis. The results were described, but not related to each other.

Response 3: Thanks for your kindly suggestion. As suggested, a general discussion was added to enhance the link of the results related to each other. In present study, the physicochemical indices and nutrients in WFV samples were analyzed firstly. The data suggested that WFV was a healthy low-sugar and low-fat food. And then the bioactive ingredients in WFV samples were investigated to further explore its functional property. In the present study, the results showed that LBP, betaine and TPC were the main bioactive components. Numerous studies have demonstrated that LBP, betaine and polyphenols are positively correlated with antioxidant activity (Yuqin Jiang, et al. Journal of Functional Foods. 2021 Feb, 77, 104340; Xiaojing Tian, er al. Biomolecules. 2019 Aug, 9(9): 389; Belhadj Slimen I et al. J Agric Food Chem. 2017 Feb, 65(4): 675-689). The antioxidant activities of WFV samples were further examined in the next experiment. The results showed that WFV exhibited superior antioxidant capacity, which was due to different raw materials and manufacturing techniques. It has been reported that the polyphenolics in vinegars mainly derive from raw materials, which are closely related to the antioxidant activity of vinegars (Qing Liu, et al. Antioxidants (Basel). 2019 Mar, 8(4): 78). Then polyphenolic compounds in WFV samples were investigated in a further experiment. p-hydroxybenzoic acid, m-hydroxycinnamic acid, and 3-(4-Hydroxy-3-methoxyphenyl) propionic acid were the main polyphenols in WFV. Moreover, the hepatoprotective effect of WFV was evaluated in vivo. The findings showed that WFV reversed the damaged hepatocytes and reduced liver damage through morphology examination. Next, some biochemical indexes were further determined to verify the hepatoprotective effect of WFV. The results suggested that WFV decreased the levels of liver function indexes, and had a positive effect on liver injury. Subsequently, the oxidative and antioxidative paraments were evaluated to explore the mechanism of hepatoprotective activity of WFV. Collectively, WFV had a role in balancing the levels of oxidation and antioxidation, which alleviated oxidative damage in the liver.

As suggested, the related contents were added in the Results and discussion section 2.1 (Page 3 Line 109-110); Results and discussion section 2.2 (Page 4 Line 138-141); Results and discussion section 2.3 (Page 4 Line 165-168); Results and discussion section 2.4 (Page 5 Line 199-200); Results and discussion section 2.5 (Page 6 Line 228-230); Results and discussion section 2.6 (Page 7 Line 259-261). Thanks again to make our manuscript more complete and logical.

Reviewer 2 Report

Manuscript ID: molecules-1781778 entitled: "Nutrition, bioactive components, and hepatoprotective activity of fruit vinegar produced from Ningxia wolfberry"

The subject undertaken in the above mentioned manuscript is very important. The authors presented a broad spectrum of analyses testing the functional properties of fruit vinegar from Ningxia wolfberry. The conducted studies are supported by promising results in the future treatment of liver disfunctions. I recommend this manuscript for publication. Please reconsider only the below suggested small corrections:

Line 85, page 2 and line 109, page 3  „indexes” or rather indices (plural)?

Line 363, page 10. Please incorporate colon „:” after the word „grouped”

Author Response

Response to Reviewer 2 Comments

Manuscript ID: molecules-1781778 entitled: "Nutrition, bioactive components, and hepatoprotective activity of fruit vinegar produced from Ningxia wolfberry"

The subject undertaken in the above mentioned manuscript is very important. The authors presented a broad spectrum of analyses testing the functional properties of fruit vinegar from Ningxia wolfberry. The conducted studies are supported by promising results in the future treatment of liver disfunctions. I recommend this manuscript for publication. Please reconsider only the below suggested small corrections:

Line 85, page 2 and line 109, page 3, “indexes” or rather indices (plural)?

Line 363, page 10. Please incorporate colon “:” after the word “grouped”

Response: Thanks for your kindly reminding. We are sorry for the mistakes. As suggested, the word “indexes” has been replaced with “indices” in Results and discussion section 2.1 (Page 2 Line 85-86), Table 1 (Page 3 Line 111), and Materials and Methods section 3.3 (Page 9 Line 318). In addition, we have incorporated colon “:” after the word “grouped” in Materials and Methods section 3.6 Animals and diets (Page 11 line 397). Thanks again for your suggestion to make our manuscript more accurate and clarified.

Reviewer 3 Report

Tian et al. reported the in vitro and in vivo study results of nutrition, bioactive compounds, and hepaprotective activity of wolfberry vinegar. Before this manuscript is suitable for publication, I would like the authors to address the following comments:

(1)   Figure 2B: how to define “disordered in arrangement”? The arrangements of cells in the control group and the experimental groups look similar. Also, how can readers tell the intactness of the cell membrane based on the pictures? Some justifications and quantitative analysis may be necessary to evaluate the healthiness of the cells.

(2)   Please specify how many batches of WFV were tested in this project. Since there is no purification step during the production of WFV, batch-to-batch variation could be a concern.

(3)   The cytotoxicity of WFV to normal cells should be discussed (control group treated with WFV).

(4)   Figure 1 seems unnecessary. Little information can be obtained by just looking at the antioxidant capacities of WFV measured by different assays. The authors may consider displaying in Figure 1 the antioxidant capacity values of apple cider and grape vinegar, to support the statement “antioxidant capability of WFV was superior than that of apple cider and grape vinegar”.

(5)   In the sentence “The antioxidant capability of WFV was superior than that of apple cider and grape vinegar, which was due to different raw materials and manufacturing techniques”, the authors need to provide either literature evidence or data before making the statement “due to different raw materials and manufacturing techniques”.

Minor points:

(6) Line 18: consider choosing another word instead of “insufficient”.

(7) Provide the definition of “TPC”, “TFC” etc. when first introducing them in section 2.2.

(8) Provide the definition of “ABTS”, “FRAP”, and “DPPH” when first introducing them in section 2.3.

(9) "Superior to" may be better than "superior than".

Author Response

Response to Reviewer 3 Comments

Tian et al. reported the in vitro and in vivo study results of nutrition, bioactive compounds, and hepaprotective activity of wolfberry vinegar. Before this manuscript is suitable for publication, I would like the authors to address the following comments:

Point 1: Figure 2B: how to define “disordered in arrangement”? The arrangements of cells in the control group and the experimental groups look similar. Also, how can readers tell the intactness of the cell membrane based on the pictures? Some justifications and quantitative analysis may be necessary to evaluate the healthiness of the cells.

Response 1: Thanks for your kindly suggestion. We are sorry for the unclear pictures of H&E staining. As suggestion, hepatic tissues were re-dyed by H&E staining to easily distinguish difference between groups. As shown in Figure 2(B), there are no obvious histopathological changes between control group and WFV group. Hepatocytes in the model group were disordered in arrangement, which were accompanied by partial inflammation, steatosis, and swelling. The hepatic cells in CCl4+WFV group and CCl4+silybin group were less steatosis and inflammation, indicating alleviation of hepatic tissue injury.

Figure 2. Effect of WFV on the damaged liver in mice. (A) Anatomical examination of liver in mice. (B) Hepatic tissues were dyed with H&E (200× magnification).

In addition, the quantitative analysis for steatosis were performed to evaluate the healthiness of the cells. As shown in Figure S1, the steatosis scores between control and WFV groups were not significant alteration. The steatosis score in CCl4 group was significantly higher than that in control group. On the contrary, the level of liver steatosis in CCl4+WFV group was significantly decreased compared with that in CCl4 group, which suggesting that the degree of hepatic steatosis was alleviated by WFV pretreatment. Thanks again for your suggestion to make our manuscript more detailed and clarified.

Figure S1. Liver stestosis score of effect of WFV on the damaged liver in mice.

As suggest, the related contents were added in Results and Discussion section 2.5 (Page 6 Line 204-255, 211-219).

Point 2: Please specify how many batches of WFV were tested in this project. Since there is no purification step during the production of WFV, batch-to-batch variation could be a concern.

Response 2: Thanks for your kindly suggestion. In this study, the raw material of WFV is wolfberry fruit. The quantity and quality of wolfberry fruit are related to the local temperature, precipitation and sunshine duration (Bartosz Kulczyński, et al; Pol J Food Nutr Sci. 2016 Jun; 66(2): 67-75; Chenxiao Duan, et al; Sci Total Environ. 2022 Jun; 827: 154317; M Virginia Palchetti, et al; Plant Physiol Biochem. 2021 Jun; 163: 166-177.). The wolfberry qualities are different every year, which influence the stability of WFV. Therefore, the wolfberry fruits in the same year were collected to produce WFV in order to reduce batch-to-batch variation as much as possible. And 3 batches of WFV were tested in this project.

As suggested, the related contents were added in Materials and Methods section 3.2 (Page 9 Line 316).

Point 3: The cytotoxicity of WFV to normal cells should be discussed (control group treated with WFV).

Response 3: Thanks for your kindly suggestion. As suggested, we added anatomical and histomorphological examination of the liver treated by WFV. In Figure 2(A), the liver of mice in WFV group was similar to that in the control group, showing normal dark red color. As shown in Figure 2(B), the liver cells in the WFV group were orderly arrangement of emission without the presence of steatosis, which was similar to that in the control group. Generally, there are no obvious anatomical and histopathological changes between control group and WFV group. The results indicate that WFV has no cytotoxic effect to normal cells.

Figure 2. Effect of WFV on the damaged liver in mice. (A) Anatomical examination of liver in mice. (B) Hepatic tissues were dyed with H&E (200× magnification).

As suggested, the related contents were added in Results and Discussion section2.5 (Page 6 Line 219-233).

Point 4: Figure 1 seems unnecessary. Little information can be obtained by just looking at the antioxidant capacities of WFV measured by different assays. The authors may consider displaying in Figure 1 the antioxidant capacity values of apple cider and grape vinegar, to support the statement “antioxidant capability of WFV was superior than that of apple cider and grape vinegar”.

Response 4: This is a good suggestion. As suggested, the antioxidant capacity values of apple cider vinegar and grape vinegar were added in Figure 1 according to the literatures. As displayed in Figure 1, the antioxidant activities of WFV were 20.842±0.644 mM Trolox/L (ABTS), and 26.736±1.238 mM Trolox/L (DPPH), respectively. Kelebek et al. [34] reported eight different brands of apple cider vinegar and grape vinegar from Turkish market. The highest antioxidant activity of apple cider vinegar was 20.19 ± 0.41 mM Trolox/L (ABTS) and 14.69±0.30 mM Trolox/L (DPPH), respectively. The highest antioxidant activity of grape vinegar was 17.96±1.34 mM Trolox/L (ABTS) and 14.43±0.97 mM Trolox/L (DPPH), respectively. By comparison, the ABTS of WFV in this study was slightly higher than that of apple cider vinegar (p> 0.05), and significantly higher than that of grape vinegar (p< 0.05). In addition, the DPPH value of WFV was significantly higher than that of two commercial fruit vinegars (p< 0.001). It has been reported that various fruits contain different bioactive substances such as polyphenols and vitamins providing antioxidant activity, which influence the properties of vinegar (Luz María Luzón-Quintana, et al. Foods. 2021 Mar; 10: 945). In addition, new bioactive compounds such as organic acids and polyphenols can be produced during the fermentation process (Roberto Mandrioli, et al. 2013 May; 405: 7941-7956; Driss Ousaaid, et al. Molecules. 2021 Dec; 27: 222). Some studies have demonstrated that the antioxidant activities of fruit vinegars were related to raw materials and manufacturing techniques (Joanna Kawa-Rygielska, et al; Molecules. 2018 Feb; 23(2): 379; C. Ubeda, et al; LWT - Food Science and Technology, 2013 Jul; 52(2): 139-145; C. Ubeda, et al; LWT - Food Science and Technology, 2011 Sep; 44(7): 1591-1596;). Generally, the results showed that the antioxidant capability of WFV was superior to that of apple cider and grape vinegar, which was due to different raw materials and manufacturing techniques. And the polyphenolics of antioxidant substances in WFV samples were investigated in further experiment.

The related contents were added in Results and Discussion section 2.3 (Page 4 Line 145-168).

Figure 1. Antioxidant capacities of WFV, apple cider vinegar, and grape vinegar [34] were measured by ABTS, DPPH assays.

Point 5: In the sentence “The antioxidant capability of WFV was superior than that of apple cider and grape vinegar, which was due to different raw materials and manufacturing techniques”, the authors need to provide either literature evidence or data before making the statement “due to different raw materials and manufacturing techniques”.

Response 5: Thanks for your information and suggestion. As suggest, we have added data comparison and literature evidence to support the statement. By comparison, the ABTS of WFV in this study was slightly higher than that of apple cider vinegar (p> 0.05), and significantly higher than that of grape vinegar (p< 0.05). In addition, the DPPH value of WFV was significantly higher than that of two commercial fruit vinegars (p< 0.001). It has been reported that various fruits contain different bioactive substances such as polyphenols and vitamins providing antioxidant activity, which influence the properties of vinegar (Luz María Luzón-Quintana, et al. Foods. 2021 Mar; 10: 945). In addition, new bioactive compounds such as organic acids and polyphenols can be produced during the fermentation process (Roberto Mandrioli, et al. 2013 May; 405: 7941-7956; Driss Ousaaid, et al. Molecules. 2021 Dec; 27: 222). Some studies have demonstrated that the antioxidant activities of fruit vinegars were related to raw materials and manufacturing techniques (Joanna Kawa-Rygielska, et al; Molecules. 2018 Feb; 23(2): 379; C. Ubeda, et al; LWT - Food Science and Technology, 2013 Jul; 52(2): 139-145; C. Ubeda, et al; LWT - Food Science and Technology, 2011 Sep; 44(7): 1591-1596). Generally, the results showed that the antioxidant capability of WFV was superior to that of apple cider and grape vinegar, which was due to different raw materials and manufacturing techniques.

As suggested, the related contents were added in Results and Discussion section 2.3 (Page 4 Line 147-164).

Minor points:

Point 6: Line 18: consider choosing another word instead of “insufficient”.

Response 6: Thanks for your kindly reminder. We are sorry for the unclear description. As suggested, we have replaced “insufficient” with “needs to be improved” in Abstract (Page 1 Line 18) of our revised manuscript.

Point 7: Provide the definition of “TPC”, “TFC” etc. when first introducing them in section 2.2.

Response 7: Thanks for your kindly reminder. As suggested, the definition of “TPC”, “TFC” has been added in Results and Discussion section 2.2 (Page 3 Line 117-118).

Point 8: Provide the definition of “ABTS”, “FRAP”, and “DPPH” when first introducing them in section 2.3.

Response 8: Thank you for your kindly reminder. As suggested, we have added the definition of “ABTS”, “FRAP”, and “DPPH” in Results and Discussion section 2.3 (Page 4 Line 145-146).

Point 9: "Superior to" may be better than "superior than".

Response 9: Thanks for your kindly suggestion. As suggested, “superior to” has replaced with “superior than” in Results and Discussion section 2.3 (Page 4 Line 163) of our revised manuscript.

Round 2

Reviewer 3 Report

The authors have addressed my comments adequately, and the revised version of the manuscript is much improved. I would recommend this manuscript to be published in Molecules, while the authors could pay some attention to my additional comments below:

(1)  Figure 2B: I appreciate the authors’ efforts in re-dying the cells. The updated Figure 2B now clearly shows the morphology of the cells in different experimental and control groups. Still, I am not convinced that one can easily tell the “arrangement” of the cells even in the new Figure 2B. I would suggest avoiding commenting on the ordering or arrangement of the cells. I am fine with the authors commenting on the “inflammation, steatosis, and swelling” of the cells based on the updated Figure 2B.

I appreciate that the authors have included Figure S1, the quantitative analysis for steatosis in the revised version. Since Figure S1 will support the arguments made based on Figure 2B, I strongly suggest the authors call out Figure S1 somewhere in the main text.

(2)  Thanks for the clarifications of the batches of the wolfberry used. Besides mentioning “three batches” (line 316), I suggest adding “All three batches of the wolfberry fruits were collected in the same year to produce WFV in order to reduce batch-to-batch variation”.

(3)  The authors have now addressed the cytotoxicity of WFV to normal cells in several places in the revised manuscript.

(4)  Figure 1 is much improved with the comparison of the antioxidant capacity values of apple cider vs grape vinegar vs WFV.

(5)  In the revised manuscript, the authors have provided satisfactory literature evidence to support the statement “The antioxidant capability of WFV was superior to that of apple cider and grape vinegar, which was due to different raw materials and manufacturing techniques”.

Author Response

Response to Reviewer 3 Comments

The authors have addressed my comments adequately, and the revised version of the manuscript is much improved. I would recommend this manuscript to be published in Molecules, while the authors could pay some attention to my additional comments below:

Point 1: Figure 2B: I appreciate the authors’ efforts in re-dying the cells. The updated Figure 2B now clearly shows the morphology of the cells in different experimental and control groups. Still, I am not convinced that one can easily tell the “arrangement” of the cells even in the new Figure 2B. I would suggest avoiding commenting on the ordering or arrangement of the cells. I am fine with the authors commenting on the “inflammation, steatosis, and swelling” of the cells based on the updated Figure 2B.

I appreciate that the authors have included Figure S1, the quantitative analysis for steatosis in the revised version. Since Figure S1 will support the arguments made based on Figure 2B, I strongly suggest the authors call out Figure S1 somewhere in the main text.

Response 1: Thanks for your kindly suggestion. As suggested, we have deleted the comments on the ordering or arrangement of the cells. In addition, Figure S1 has been added in Figure 2C. Figure 2 was shown as follows:

Figure 2. Effect of WFV on the damaged liver in mice. (A) Anatomical examination of liver in mice. (B) Hepatic tissues were dyed with H&E (200× magnification). (C) Steatosis score of liver in mice.

The related contents were revised in Results and Discussion section 2.5 (Page 6 Line 212; 216), Figure2 and Figure legend (Page 6 Line 231; 233).

Point 2: Thanks for the clarifications of the batches of the wolfberry used. Besides mentioning “three batches” (line 316), I suggest adding “All three batches of the wolfberry fruits were collected in the same year to produce WFV in order to reduce batch-to-batch variation”.

Response 2: Thanks for your kindly suggestion. As suggested, the related contents were added as follows: Three batches of WFV were reserved at 4 °C for subsequent analysis. All three batches of the wolfberry fruits were collected in the same year to produce WFV in order to reduce batch-to-batch variation. Thanks again for your suggestion to make our manuscript more detailed and clarified.

The related contents were revised in Materials and Methods section 3.2 (Page 9 Line 316-317).
